# HISFCOS: Half-Inverted Stage Block for Efficient Object Detection Based on Deep Learning

**DOI:** 10.3390/jimaging8040117

**Published:** 2022-04-17

**Authors:** Beomyeon Hwang, Sanghun Lee, Seunghyun Lee

**Affiliations:** 1Department of Plasma Bio Display, Kwangwoon University, Seoul 01897, Korea; clown1320@kw.ac.kr; 2Ingenium College of Liberal Arts, Kwangwoon University, Seoul 01897, Korea; shlee@kw.ac.kr

**Keywords:** FCOS, HIS block, object detection, CNN, deep learning, feature pyramid

## Abstract

Recent advances in object detection play a key role in various industrial applications. However, a fully convolutional one-stage detector (FCOS), a conventional object detection method, has low detection accuracy given the calculation cost. Thus, in this study, we propose a half-inverted stage FCOS (HISFCOS) with improved detection accuracy at a computational cost comparable to FCOS based on the proposed half inverted stage (HIS) block. First, FCOS has low detection accuracy owing to low-level information loss. Therefore, an HIS block that minimizes feature loss by extracting spatial and channel information in parallel is proposed. Second, detection accuracy was improved by reconstructing the feature pyramid on the basis of the proposed block and improving the low-level information. Lastly, the improved detection head structure reduced the computational cost and amount compared to the conventional method. Through experiments, the proposed method defined the optimal HISFCOS parameters and evaluated several datasets for fair comparison. The HISFCOS was trained and evaluated using the PASCAL VOC and MSCOCO2017 datasets. Additionally, the average precision (AP) was used as an evaluation index to quantitatively evaluate detection performance. As a result of the experiment, the parameters were increased by 0.5 M compared to the conventional method, but the detection accuracy was improved by 3.0 AP and 1.5 AP in the PASCAL VOC and MSCOCO datasets, respectively. in addition, an ablation study was conducted, and the results for the proposed block and detection head were analyzed.

## 1. Introduction

Recently, computer vision has been studied in related applications, such as object detection [1,2], semantic segmentation [3,4], and super resolution [5], because of the development of convolutional neural networks (CNNs). Object detection is a fundamental task in computer vision that classifies the categories of objects after a regression process to determine the location of an object. It is a core technology in autonomous driving [6], monitoring systems [7], and face recognition [8].

Deep-learning-based object detection methods are classified into one- and two-stage methods. It is divided into anchor-based and anchor-free methods according to the method for defining the anchor in detail. The two-stage method uses a region proposal network (RPN) to find regions where objects are most likely to be present. On the basis of the selected region, classification is performed after localization. Representative methods include region-based convolutional neural networks (R-CNNs) [9,10,11]. Unlike the two-stage method, the one-stage method simultaneously handles both localization and classification. Compared to the two-stage method, there is a trade-off between detection accuracy and detection speed. The one-stage method has the advantage of detection speed. Representative methods include the single shot multibox detector (SSD) [12] and you only look once (YOLO) [13,14]. Additionally, the one-stage method improves detection accuracy by extending the depth of the model to improve low accuracy and the feature pyramid [15] structure, which uses feature maps of different scales.

Object detection researchers are studying anchors that affect detection performance. An anchor-free method for predicting the class of an object and learning its location, similar to a fully convolutional network (FCN) [16], was studied rather than defining an anchor as in a conventional detector. The fully convolutional one-stage detector (FCOS) [17] is an example of a representative method.

In this paper, we propose HISFCOS with improved detection accuracy while retaining FCOS network complexity. In the conducted work, the accuracy of the proposed network was improved using three main modules (half-inverted stage (HIS) block, HIS feature pyramid, and lightweight detection head). First, the HIS block improved detection accuracy by minimizing feature loss through parallel operation for the proposed spatial and channel information. Second, low-level feature information loss caused by the conventional method was minimized by reconstructing the feature pyramid on the basis of the proposed block. Lastly, the amount of computation was minimized while maintaining accuracy by improving the conventional detection head structure. The proposed method showed high detection accuracy in experiments with similar computational costs comparable to the FCOS. Through an ablation study, we analyzed the contribution of the detection head structure and the HIS block in HISFCOS.

The main contributions of this work are as follows:We propose an HIS block that reduces loss of spatial and channel information.We improved detection accuracy by reconstructing the feature pyramid on the basis of the proposed block and improving the low-level information.We propose a lightweight detection head that reduces the amount of computation by improving the structure of the conventional detection head.

## 2. Related Works

### 2.1. Fully Convolutional One-Stage Detector

The FCOS is an anchor-free, one-stage detector network with a feature pyramid structure that uses ResNet [18] as its backbone network. Figure 1 shows the FCOS structure. This method was proposed to address the drawbacks of the conventional anchor-based method. The anchor-based method affected the detection performance according to the box design, such as the size, aspect ratio, and number of predefined anchors. Additionally, many bounding boxes are created to achieve high recall. A recall is the ratio of object detection when object detection is more important than precision. However, class-imbalance problems arise because many bounding boxes are assigned to the negative samples. Therefore, FCOS predicts the class of an object in pixel units, such as an FCN, without using an anchor. When the predicted sample is positive, a detector of the anchor-free method, which detects objects without using an anchor, is proposed using distance vector regression with the predicted box based on the center of the object. Additionally, a feature pyramid structure was used to improve the low correct recall problem caused by not using an anchor. Lastly, a centerness loss function was proposed to solve the problem of detection accuracy being reduced as a low-level bounding box with a low score moved away from the center. The centerness loss function improves detection accuracy by removing the low-level bounding box in a manner that gives weights when the distance from the center is considered in relation to the center distance of the object.

### 2.2. Depthwise Convolution

Depthwise convolution (Dwconv), unlike standard convolution, can extract the spatial information of each channel without being affected by all channels of the input image. In other words, calculations for each channel are performed in the spatial direction without the involvement of other channels. Thus, each kernel has a parameter for a single channel. Consequently, only spatial information unique to each channel can be learned, which is the same as in the special case where the number of groups in the group convolution equals the number of channels. On the basis of this structure, MobileNet [19,20] was proposed for limited environments such as embedded devices using depthwise separable convolution [21], which combines Dwconv and 1×1 convolution to exponentially reduce the amount of computation and enable real-time operation. Figure 2 shows example of depthwise convolution with kernel = 3, input tensor size 8×8 (C×H×W). The symbols *C*, *H*, *W* and ⨀ are the channel height, width and matrix product.

### 2.3. Channel Attention

Channel attention (CA) [22] is a technique that emphasizes a specific channel using the correlation between channels in the feature map. This process is illustrated in Figure 3.

First, the input feature map contains a vector representing each channel as a vector of the same size as the channel size of the feature map through global average pooling (GAP). The vector is compressed into a vector with meaningful information using a fully connected (FC) layer. Nonlinearity is added to the compressed vector using a rectified linear unit (ReLU) activation function. The vector is compressed, and nonlinearity is added via the second FC layer. A compressed vector is generated using the sigmoid activation function to enhance the vector of the channel size with a value between zero and one. The emphasized vector is refined using the input feature map and element multiplication operation to refine the unnecessary feature map in each channel and generate the emphasized feature for the object in the channel. The channel attention is expressed using Equation (Equation 1):(1)$$CA=\;{{(\sigma }_{2}{(FC}_{2}{(\sigma }_{1}(FC}_{1}\left(GAP\left(F_{Input}\right)\right)))))\cdot F_{Input}$$
where $F_{Input}$, $\sigma_{1}$, $\sigma_{2}$, and GAP are input feature map, ReLU, sigmoid, and global average pooling, respectively.

## 3. Proposed Method

This section describes the proposed method. HISFCOS consists of a backbone network, HIS feature pyramid, and a lightweight detection head.

Figure 4 shows the proposed HISFCOS architecture. First, ResNet-50, the same as the conventional FCOS, was used as the backbone network. The feature map extracted from the backbone network was used as an input for the reconstructed feature pyramid on the basis of the HIS block that minimizes feature loss. In addition, a bottom–up path was added to the feature pyramid of the top–down path to improve the lack of low-level information in the conventional network. As a result, feature information is improved by combining the high-level feature map and the low-level information. Lastly, unlike the conventional detection head composed of standard convolution, an inverted residual block structure is applied to reduce the amount of computation. The proposed method detects large and small objects for each scale using five heads that had passed through a feature pyramid. Details are covered in Section 3.1, Section 3.2 and Section 3.3.

### 3.1. Pyramid with Half-Inverted Stage Block

#### 3.1.1. Half-Inverted Stage Block

Feature information in the FCOS feature pyramid is lost in the deep layer. Therefore, we propose an HIS block that reduces feature loss by simultaneously calculating spatial and channel information. Figure 5 shows the structure of the HIS block.

The structure of the proposed HIS block is as follows. First, if a large channel is used, unnecessary features and computations are increased. Therefore, the input feature map is compressed to $\frac{1}{2}$ channel size using a 1×1 convolution. Subsequently, to minimize the loss of spatial and channel information, the optimized operation for each operation was used in parallel. Dwconv, which extracts spatial information through spatial operations in channel units, and CA, which emphasizes important features such as objects in the channel space, are used. (Channel attention replaces the FC layer with a 1×1 convolution to prevent an increase in the amount of computation). To refine the feature information extracted by a parallel operation, it is first combined with a concatenation operation. Feature loss generated during the compression process was minimized by combining the compressed and refined features with the size of the first input channel $\frac{1}{2}$. Lastly, to improve the spatial information required for object detection, spatial information was improved using a dilated convolution with a wide receptive field and a computational cost comparable to the standard convolution. Equations (Equation 2) and (Equation 3) represent the spatial and channel feature combinations and refinement process with the HIS block, respectively.
(2)$$M_{Feature\;connection}\left(x\right)={Conv}_{3\times 3}\left(concat\left[Dwconv\left(x\right);\;CA\left(x\right)\right]\right)$$
(3)$$M_{HIS}\left(x\right)={Dilated}_{3\times 3}^{r=2}\left({{concat[M}_{Feature\;connection}(Conv}_{1\times 1}\left(x\right)\right);\;{Conv}_{1\times 1}\left(x\right)])$$
where each symbol is $Dilated_{3\times 3}^{r=2}$, and concat represent a dilated convolution and concatenation operation with a kernel size of 3 and dilated ratio of 2, respectively.

#### 3.1.2. HIS Feature Pyramid

FCOS is composed of a feature pyramid with a top–down path structure. Some deep layers do not combine with low-level features to recover the loss after upsampling, resulting in feature loss. Therefore, feature loss was prevented by rebuilding the feature pyramid using the proposed HIS block. First, by incorporating a bottom–up path into the conventional top–down path structure, low-level feature loss caused by the model depth is minimized by combining low-level information such as contours and edges with high-level information such as texture and shape in each feature map. Second, because there is no process in the conventional method for restoring information loss in some features of the feature pyramid, there may be a part-false-positive problem of detecting multiple objects owing to feature loss when extracting a large object. Therefore, by combining the features of the backbone network, the loss of features is minimized, allowing for the part-false-positive problem to be solved when large objects are detected.

### 3.2. Lightweight Detection Head

A conventional detection head has a structure in which each classification and regression branch repeat 3×3 standard convolution four times for feature refinement. The conventional structure improves accuracy in a detector with a shallow structure [17,23]. However, a detection head with a structure that repeats the standard convolution reduces detection accuracy compared with the high computational cost. Therefore, the proposed structure is applied to a conventional detection head using an inverted residual block [24] structure to lower the high computational cost. The computational cost of an ablation study was compared with that of a conventional detection head. Figure 6 shows the structures of the conventional and proposed detection heads.

First, the proposed detection head extends the channel using 1×1 convolution. Spatial information was extracted from the extended feature map using 3×3 Dwconv. The extracted spatial information was compressed using 1×1 convolution. By combining the extracted spatial information with the channel axis, features could be extracted at a lower computational cost compared with standard convolution. This structure is an inverted residual block. Features were then refined using 3×3 standard convolution. By replacing it with this structure, efficient detection was possible at a lower computational cost than that of the conventional method.

### 3.3. Loss Function

The classification loss and bounding box regression loss functions are used as loss functions in the proposed method, in the same manner as the FCOS. We used focal loss [23] as the classification loss function. The cross-entropy loss function compares the proposed method with the ground truth, and outputs an error. However, when the standard cross-entropy loss function is used, the sample, which is easily detected, dominates the overall loss function. In object detection, most samples have a larger ratio of background samples to foreground samples, which causes class-imbalance problems. The use of such unbalanced samples for training is inefficient. Therefore, focal loss, which improved the standard cross-entropy loss function, was used as the classification loss function. The cross-entropy and focal loss functions are expressed using Equations (Equation 4) and (Equation 5), respectively:(4)$$CrossEntropy\left(p,y\right)=\left\{\begin{array}{cc} -\log(p), & ify=1\\ -\log(1-p), & \mathrm{otherwise}. \end{array}\right.$$
where *p* and *y* are the output values of ground truth and model, respectively.
$$p_{t}=\left\{\begin{array}{cc} p, & \mathrm{if}\;y=1\\ 1-p, & \mathrm{otherwise} \end{array}\right.$$
(5)$$FocalLoss\left(p,g\right)={-\alpha \left(1-p_{t}\right)}^{\gamma }\cdot \log\left(p_{t}\right),$$

Focal loss becomes smaller than conventional cross-entropy loss when $p_{t}$ approaches one. Conversely, as $p_{t}$ approaches zero, loss increases. Here, α,γ are hyperparameters that control the contribution to the loss function. When γ=0, it is the same as the conventional cross-entropy loss function. We use α,γ, which had the same value as FCOS α=2, γ=0.25.
(6)$$GIoU\left(B,B^{gt}\right)=1-IoU+\;\frac{\left|C-B\cup B^{gt}\right|}{\left|C\right|},$$

For the bounding box regression loss function, generalized intersection over union (GIoU) [25] was used as a regression loss function on the basis of the intersection of union (IoU) [26]. Equation (Equation 6) represents GIoU, where each symbol *B*,$B^{gt}$,and *C* represents the predicted bounding box, ground truth, and the small size area covering the predicted bounding box and ground truth, respectively.

The centerness loss function [17], which assigns weight to the bounding box with a low score away from the center, is a loss function that determines whether an object exists in the center. Therefore, binary cross-entropy (BCE) loss, a special form of cross-entropy, is used as a function to compare the two cases. Equation (Equation 7) is a centerness loss function expressed as follows:(7)$$CenternessLoss\left(Y,\hat{Y}\right)=-(Y\log\hat{Y}+\left(1-Y\right)\log\left(1-\hat{Y})\right),$$
where *Y* and $\hat{Y}$ are output values of ground truth and model, respectively.

The total loss function consists of classification loss, regression loss, and centerness loss functions. Total loss is expressed using Equation (Equation 8) as follows: (8)$${Loss}_{Total}=FocalLoss+GIoULoss+CenternessLoss.$$

## 4. Experiment Results and Discussion

### 4.1. Implementation Details

In this experiment, the public datasets PASCAL VOC [27] and MS COCO2017 [28] were used for HISFCOS performance validation and evaluation. The backbone network was pretrained on the Imagenet-1K dataset before experimentation. The hyperparameters used in the experiment were as follows: stochastic gradient descent (SGD) as used was the optimizer, momentum was set at 0.9, and weight decay was set at 5 × 10${}^{-3}$. For each dataset, batches were trained using 32 and 10 epochs, and 50 and 30 epochs were used. Additionally, for the input resolution, PASCAL VOC was used in the same manner as 512×512 and the MSCOCO dataset was used in the same manner as the conventional FCOS. An initial learning rate of 1 × 10${}^{-2}$ was used, and in the case of the PASCL VOC dataset, the learning rate decreased by 0.1 for each 2 and 2.1 K, and in the case of the MSCOCO, the learning rate decreased by 0.1 at 60 and 90 K. The data augmentation technique was as follows: random crops are randomly cropped according to the resolution of the input image. The color jitter randomly changes the brightness, saturation, and hue of the input image. Finally, a random rotation changed the height and width of the input image at random. The hardware and framework used in the experiment are listed in Table 1. The code is available at https://github.com/hby1320/pytorch_object_detection (accessed on 15 March 2022).

In this study, detection accuracy was evaluated using the average precision (AP) [29], which is a dataset evaluation metric used in the field of object detection. The AP was calculated by averaging the maximal precision of the recall value on the precision–recall (PR) curve. Precision and recall were obtained using Equations (Equation 9) and (Equation 10), respectively, as follows: (9)$$Precision\;=\;\frac{TP}{TP+FP},$$
(10)$$Recall\;=\;\frac{TP}{TP+FN},$$

When calculating *AP*, if there is only one classification class, it is defined as Equation (Equation 11).
(11)$$AP\;=\frac{1}{11}\sum_{r\in 0.0,\;\ldots ,1.0}\rho_{interp}\left(r\right),$$

Recall levels 0.0, …, 1.0 calculated the *AP* of each class as the mean of the maximal precision value for 11 levels. it is necessary to calculate the mean value for the *AP* because the classification task in the public dataset has more than each class. Equation (Equation 12) defined the average of *AP* for all classes.
(12)$$mAP\;=\frac{1}{N}\sum_{i=1}^{N}{AP}_{i},$$
where TP, FP, FN, *r*, $\rho_{interp}$, *N*, and $AP_{i}$ denote the true positive, false positive, false negative, recall, precision value of each recall, total number of classes, and *AP* value of $i_{th}$ class, respectively.

### 4.2. Comparison of Other Networks

#### 4.2.1. Dataset

In this study, training and tests were conducted using object detection datasets PASCAL VOC (07+12) and MSCOCO 2017. PASCAL VOC was trained and evaluated. When classified into a total of 20 classification categories, it was divided into 8324 train datasets, 11,227 validation datasets, and 4952 test datasets were used. In this study, evaluation and resection studies were conducted using 4952 tests and datasets after learning using the training and evaluation dataset. In addition, MSCOCO dataset has a total of 80 classification categories and consisted of 118,287 train datasets, 5000 validation datasets, and 4952 evaluation datasets. In this study, it was used for comparison with the conventional network and other networks.

#### 4.2.2. PASCAL VOC

We compared other networks with HISFCOS using the PASCAL VOC dataset, which is widely used in object detection. At this time, a fair comparison was carried out using input resolution of 512×512, which is frequently used in the one-stage detector in the PASCAL VOC dataset.

Table 2 shows the results of comparing the detection accuracy (mAP) and number of parameters with those of other networks in the PASCAL VOC 2007 test dataset. The proposed method has parameters similar to those of the conventional method, and detection accuracy was improved by approximately 3.0%. Additionally, compared with R-FCN, which is a two-stage detector more specialized in detection accuracy than in speed, detection accuracy of approximately 0.9% was achieved. This demonstrates the usefulness of the proposed HIS block and lightweight detection head.

Figure 7a shows that FCOS is difficult to detect overlapping objects due to spatial and low-level feature information loss during feature extraction. In addition, the bottle class at the bottom of Figure 7 is difficult to detect when there is a feature loss. The proposed method uses HIS blocks to reduce spatial and channel loss. In addition, by reconstructing the feature pyramid structure, low-level information was improved to improve the detection performance of difficult-to-detect overlapping objects. In Figure 7b, compared to the conventional method, the proposed method showed improved detection results for objects that are difficult to detect such as overlapping objects.

#### 4.2.3. MSCOCO

Unlike the PASCAL VOC dataset, the MSCOCO dataset was evaluated using the average of values between 50% and 95% as the IoU threshold. Table 3 shows a comparison of detection performance between the MSCOCO 2017 minival dataset method and other object detection methods. HISFCOS was tested with the same input resolution of 800×1333 pixels as the FCOS. The proposed method achieved a detection accuracy of approximately 38.9% using ResNet-50 as the backbone network in the MSCOCO dataset. Compared with FCOS, detection accuracy was improved by approximately 1.5%. Additionally, compared with the two-stage method, the proposed method showed similar detection accuracy. Compared with MaskR-CNN, which is a two-stage method, the proposed method has approximately four times fewer parameters and 0.5% higher detection accuracy. Figure 8 shows the detection results compared with the conventional method for the MSCOCO dataset.

Figure 8a shows the top of the image, where false detection for the knife class was confirmed. Owing to the loss of low-level information in the conventional method, similar feature information was misidentified when detecting an object, and a knife class that did not exist in the input image was erroneously detected. Additionally, the lower image in Figure 8a was a part-false-positive problem for a training class with a large aspect ratio. In the conventional method, some object information was lost during the upsample process in the feature pyramid, resulting in the incorrect detection of a large object. However, because the proposed method minimized the loss of features, it was possible to accurately detect it without false detection, as shown in Figure 8b.

### 4.3. Ablation Study

#### 4.3.1. Evaluation HISblock Analysis

First, to validate the effectiveness of the proposed method, an experiment was conducted using the PASCAL VOC 2007 test dataset. Table 4 shows the detection accuracy between each class of HISFCOS and FCOS in the PASCAL VOC 07 test dataset. The proposed method improved detection accuracy for all objects compared with the conventional method. Detection accuracy was improved by minimizing the feature loss by reconstructing the feature pyramid structure on the basis of the proposed HIS block. Additionally, the detection performance for the table, bottle, plant, and chair classes was significantly improved, which are difficult to detect because they overlap with other objects in the conventional method. Furthermore, in the case of objects with a large aspect ratio, such as trains and sofas, the conventional method has the problem of part-false-positive sample detection. However, HISFCOS minimizes the loss in each feature map through feature pyramid reconstruction based on the HIS block. The improvement in detection accuracy for previously difficult-to-detect objects was confirmed by resection studies.

#### 4.3.2. Lightweight Detection Head Analysis

The structure of the proposed detection head was compared with that of the FCOS. Table 5 presents a comparison of the structure of the proposed detection head. The base network is based on a feature pyramid reconstructed with the proposed HIS block. When the conventional detection head was used, detection accuracy was reduced by approximately 0.7%. In shallow structures such as the FCOS structure, the process of refining unnecessary features is insufficient. Therefore, stagnating the features using convolution iterations in the detection head is effective. However, as the network deepens in sufficiently refined features, unnecessary operations that repeat many convolutions simply increase the computational cost. Furthermore, it was confirmed that the detection accuracy was reduced. Most object detection networks have a structure in which the standard convolution is not repeated more than twice in the detection head of the network. Therefore, the proposed method is an efficient detection head structure that can improve accuracy while lowering computational cost.

## 5. Conclusions

In this study, we proposed HISFCOS, an efficient object detection network that achieves high accuracy while maintaining a computational cost comparable to that of FCOS. First, the proposed method had high prediction accuracy. On the basis of the HIS block, feature loss that occurs during feature extraction was reduced, and spatial information was improved. By rebuilding the feature pyramid on the basis of the proposed block, detection accuracy was improved by minimizing feature loss in the FCOS and improving the low-level information required for each feature map. Second, by improving the structure of the detection head, the calculation cost was maintained while accuracy was improved. By applying the inverted bottleneck structure to the section where the internal convolution of the conventional detection head structure was repeated, accuracy was improved while maintaining comparable computational cost to that of the conventional method. The proposed method showed high detection accuracy with similar computational cost compared with the conventional detection results because of experiments on the PASCAL VOC and MSCOCO2017 datasets. In the future, we plan to conduct weight reduction research that can be applied to any industrial application field.

## Figures and Tables

**Figure 1 jimaging-08-00117-f001:**
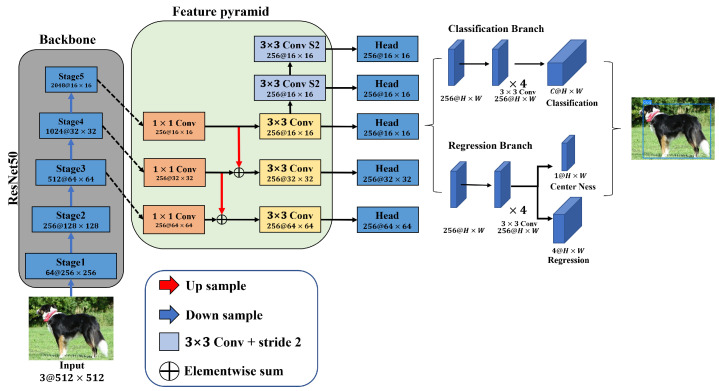
Structure of FCOS. It uses ResNet-50 as the backbone and is a simple object detector using a feature pyramid structure for multiscale detection.

**Figure 2 jimaging-08-00117-f002:**
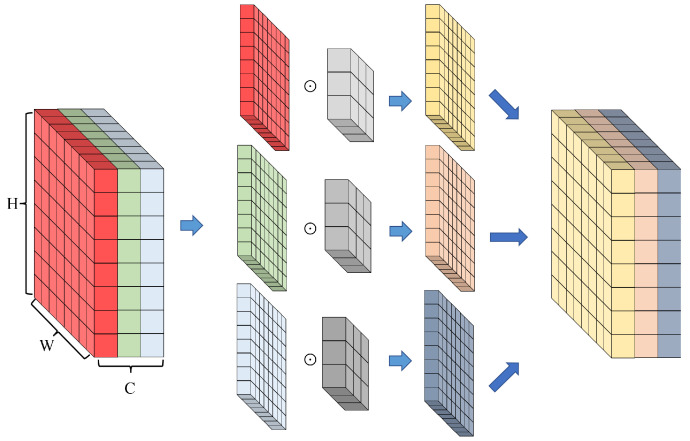
Depthwise convolution diagram. Convolution kernel separated for each channel, a 2D kernel of 3×3 is attached to each separated matrix of size 3×8×8, each convolution operation is performed, and then only channel wise spatial information is learned through recombination.

**Figure 3 jimaging-08-00117-f003:**
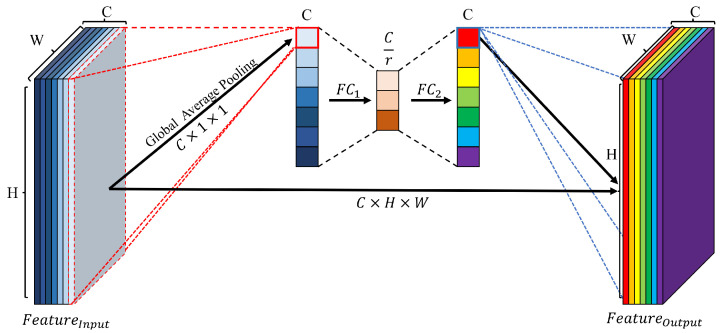
Architecture of channel attention. Channel attention compresses and readjusts the channels, attention to the features of each channel. where C, H, W and r denote the channel, height, width and channel reduction ratio, respectively.

**Figure 4 jimaging-08-00117-f004:**
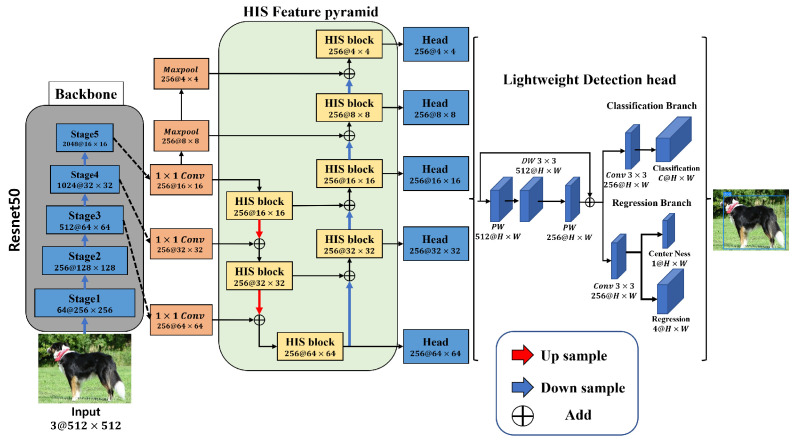
Proposed method architecture We reduced feature loss by rebuilding the feature pyramid on the basis of the proposed block. In addition, the increase in computational cost is minimized through the proposed detection head. Normalization and activation layers are omitted for simplicity.

**Figure 5 jimaging-08-00117-f005:**
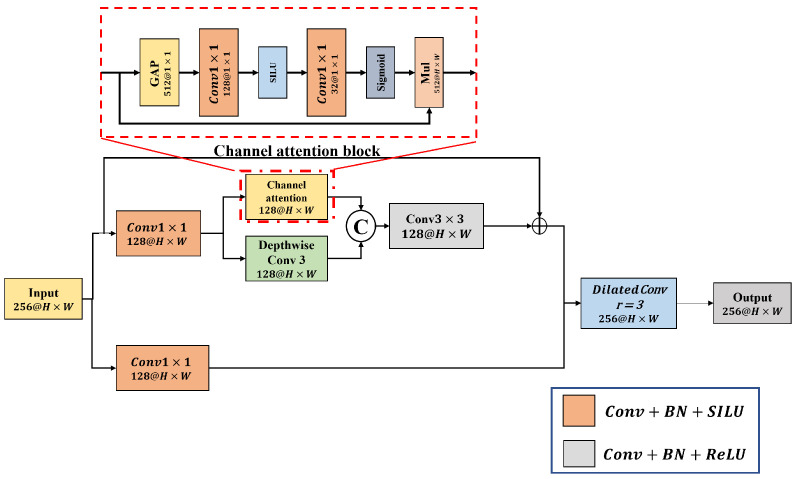
Structure of rebellion stage block. HIS block computes spatial and channel information in parallel to reduce loss of feature, where GAP, BN, ReLU, and SiLU represent the global mean pooling, batch normalization, modified linear unit, and sigmoid linear unit activation functions, respectively.

**Figure 6 jimaging-08-00117-f006:**
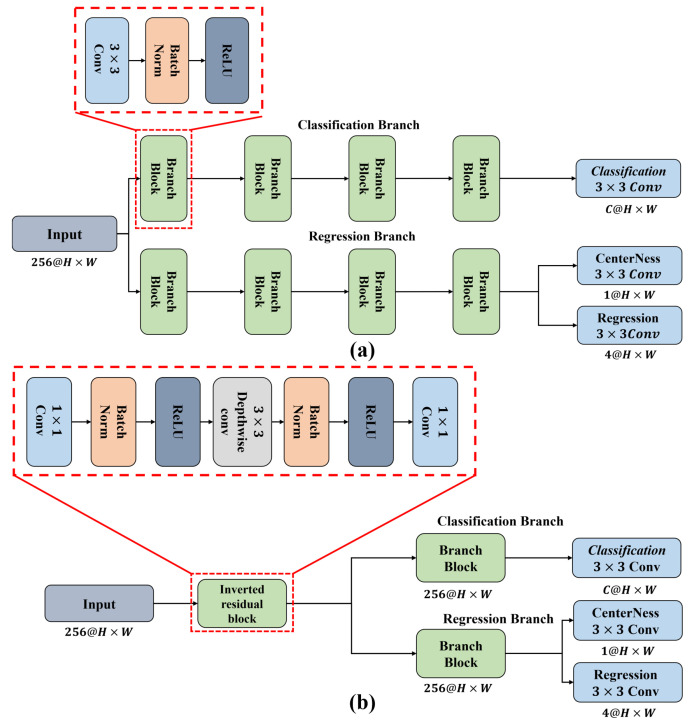
Detection head architecture: (**a**) Conventional decoupled head. It introduces repeated 3×3 convolutions to extract the feature for each task. (**b**) Proposed detection head. It used an inverted residual block, it was possible to extract features for each task with fewer operations and parameters shared than those of the conventional method.

**Figure 7 jimaging-08-00117-f007:**
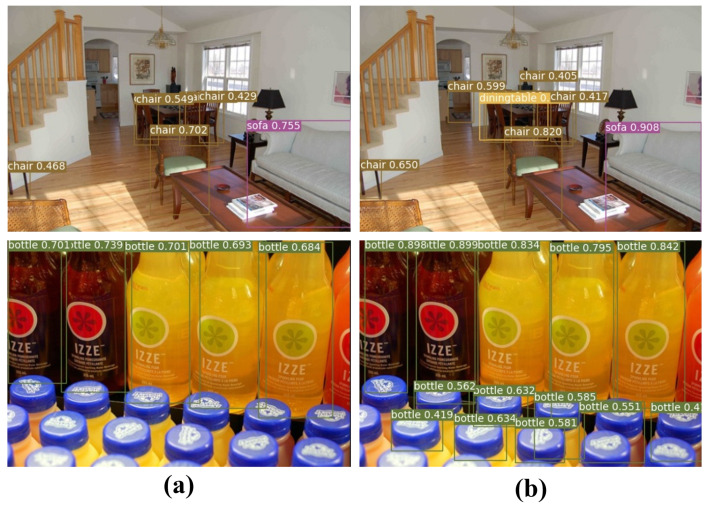
Comparison of detection results of the PASCAL VOC 2007 test dataset. (**a**) FCOS and (**b**) proposed HISFCOS method can effectively detect overlapping objects.

**Figure 8 jimaging-08-00117-f008:**
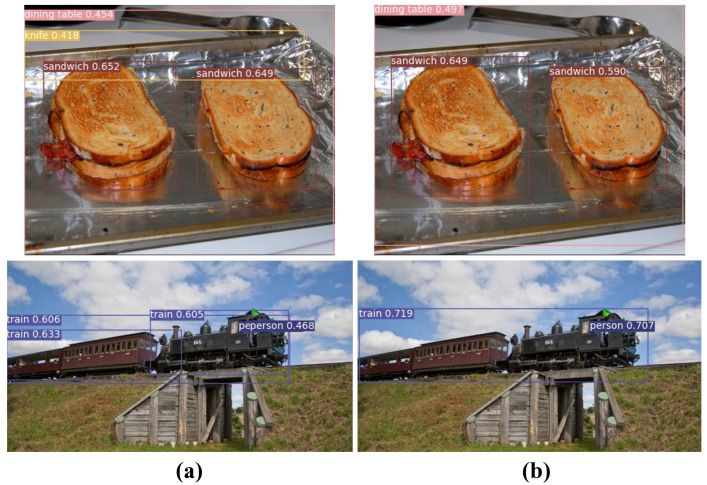
Comparison of detection results of MSCOCO 2017 minival dataset. (**a**) FCOS and (**b**) HISFCOS. As in the image below, the proposed method could effectively detect large objects.

**Table 1 jimaging-08-00117-t001:** Hardware and software environment.

Items	Descriptions
CPU	AMD Ryzen 3700X
GPU	NVIDIA RTX 3090 24 GB
RAM	64 GB
OS	Ubuntu 21.04
Framework	Pytorch 1.10

**Table 2 jimaging-08-00117-t002:** Comparison of other networks with PASCALVOC 07 test dataset.

Networks	Backbone	Input Resolution	Params (M)	mAP (%)
**Two-stage**				
Fast R-CNN [9]	VGG-16	600×1000	-	70.0
Faster R-CNN [10]	VGG-16	600×1000	134.7	73.2
OHEM [30]	VGG-16	600×1000	-	74.6
R-FCN [31]	ResNet-101	600×1000	50.9	80.5
CoupleNet [32]	ResNet-101	600×1000	-	82.7
**One-stage**				
SSD300 [12]	VGG-16	300×300	26.3	74.1
SSD512 [12]	VGG-16	512×512	29.4	76.0
YOLOv2 [14]	DarkNet-19	544×544	51.0	78.6
YOLOv3+mixip [33]	DarkNet-53	416×416	65.2	83.6
FCOS	ResNet-50	512×512	32.1	78.4
HISFCOS(our)	ResNet-50	512×512	32.6	81.4

**Table 3 jimaging-08-00117-t003:** Comparison of other networks using the MSCOCO 2017 minival dataset.

Networks	Backbone	Input Resolution	Params (M)	AP (%)	AP50 (%)	AP75 (%)
**Two-stage**						
CoupleNet [32]	ResNet-101	800×1024	-	34.4	54.8	37.2
FasterR-CNN [34]	ResNet-50	800×1024	39.8	36.7	57.3	39.3
MaskR-CNN + GRoIE [35]	ResNet-50	800×1333	-	38.4	59.9	41.7
**One-stage**						
YOLOv3 [36]	DarkNet-53	608×608	65.2	33.0	57.9	34.4
RetiaNet + Foveabox [37]	ResNet-50	800×1333	-	36.4	56.2	38.7
FSAF [38]	ResNet-50	800×1333	-	37.2	57.2	39.4
FCOS [17]	ResNet-50	800×1333	32.1	37.4	56.1	40.3
HISFCOS(our)	ResNet-50	512×512	32.6	34.0	51.8	36.1
HISFCOS(our)	ResNet-50	800×1333	32.6	38.9	57.4	41.9

**Table 4 jimaging-08-00117-t004:** Ablation study for HISFCOS analysis on PASCAL VOC 2007 test dataset.

FCOS	aero	bike	bird	boat	bottle	bus	car	cat	chair	cow
	83.1	85.4	81.3	72.4	60.6	83.2	87.6	91.7	57.8	81.8
	**table**	**dog**	**horse**	**mbike**	**person**	**plant**	**sheep**	**sofa**	**train**	**tv**
	64.5	88.1	87.0	82.3	83.8	53.8	81.9	73.7	87.9	79.4
HISFCOS (w/o Lightweight detection head)	**aero**	**bike**	**bird**	**boat**	**bottle**	**bus**	**car**	**cat**	**chair**	**cow**
	81.0	87.3	84.5	74.6	66.8	85.3	88.7	93.4	60.9	82.8
	**table**	**dog**	**horse**	**mbike**	**person**	**plant**	**sheep**	**sofa**	**train**	**tv**
	68.7	90.6	87.5	87.2	84.9	55.4	83.1	77.0	90.2	79.8
**HISFCOS**	**aero**	**bike**	**bird**	**boat**	**bottle**	**bus**	**car**	**cat**	**chair**	**cow**
	85.4	88.6	83.8	76.1	65.7	88.2	89.0	93.3	58.9	84.8
	**table**	**dog**	**horse**	**mbike**	**person**	**plant**	**sheep**	**sofa**	**train**	**tv**
	72.4	89.9	90.0	86.9	85.1	56.3	85.2	74.1	91.3	81.6

**Table 5 jimaging-08-00117-t005:** Ablation study for detection head analysis on the PASCAL VOC 2007 test dataset.

Method	Params (M)	GFLOPS (G)	mAP (%)
FCOS	32.1	103.1	78.4
Detection head (Covn ×4)	35.9	117.6	80.7
Detection head (Covn ×2)	33.5	92.0	81.0
Detection head (Proposed)	32.6	82.0	81.4

## Data Availability

Data presented in this study are openly available online: http://host.robots.ox.ac.uk/pascal/VOC/voc2007/index.html, http://host.robots.ox.ac.uk/pascal/VOC/voc2012/index.html (accessed on 26 May 2020), reference number [27], and https://cocodataset.org/#home (accessed on 26 May 2020), reference number [28].

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
