# Peer review of "HISFCOS: Half-Inverted Stage Block for Efficient Object Detection Based on Deep Learning"

_2313-433X, 2022, doi:10.3390/jimaging8040117_

Round 1

Reviewer 1 Report

The authors propose an improvement of the FCOS Network for object detection. The modification proposed by the authors lies in 2 changes: the introduction of a Half-Inverted block to prevent loosing low-level feature information, and a lightweight detection head. The paper is technically sound and the experiments conducted on the PASCAL VOC and COCO datasets tend to show that the modification improves significantly the base network. Also, the code is available, seems to do what it is said in the paper (had a quick look on it but did not run it properly) and so, the results look reproducible. However, I have some concerns about some assertions that are not always justified.
1. It is said that the low-level feature information is lost in the FCOS pyramid of the base network. In sec. 3, it is then asserted that the HIS block can recover this information. However, it seems that the base network already combines low-level features with up-sampled deeper features. Could you please be more specific about what is lost by FCOS ? The HIS blocks are more evolved than blocks used in the original FPN, and it is not obvious to me that the performance gain is due to the reason asserted by the authors. 2. The motivation for the lightweight detection head is related to computing performance. However, the introduction of the HIS blocks looks more demanding in computing facilities (Table 2 reports 32.6M vs 32.1M weights). In sec. 4, only the weight of the detection head is shown whereas the weight of the whole network is more meaningful to use HISFCOS on compute-limited devices. It should be stated that HISFCOS is heavier than the base network even with the new detection heads. 3. The ablation study in sec. 4 mix the modifications HIS blocks + head together, it would have been better to see the performance of the HIS blocks alone first, and then with the proposed detection heads.
# Typos
2.56: Minimizes → reduces 2.59-61: "that reduces the amount of computation while maintaining the computational cost" → unclear and misleading 2.67 affected → affects 5.114 minimizes → reduces 6.142 ~~each symbol~~ ... represent~~ing~~ 6.147 feature loss was ~~improved~~ *prevented* 6.166 Figure 6 leaks on the left margin 7.185 Entorpy → Entropy 7.187 g → y + missing conditions in the case for pâ‚œ definition 7.194 [[25]] (extra brackets) 7.202 what are Υ and Υ' ? 10.261-263 to detect [WHAT?] because 10.280 Table 3 too large 11.304 Table 4 too large 11.305 Capitalize (or lowercase) title 12.318 Table 5 too large

Author Response

Dear reviewer

I hope this review answer finds you well.

I have edited the contents of the review.

Thank you for your review and generous comments.

The best regards authors

Reviewer 2 Report

First and foremost, I'd like to express my appreciation to all of the authors for their efforts in producing this paper. This paper is well-structured and contains adequate research content, in my opinion. Kindly see the following for my feedback on this paper.

Referring to Figure 1. (i) It is strongly advised to avoid placing the figure before the first mention of it in the text. Do amend the other figures if necessary. (ii) The red and blue arrows in Figure 1 can be made thicker to better distinguish themselves from the black arrow. (iii) It's a good idea to include the sampling rate right next to the Up and Down sample legend/label. (iv) Define FPN, PW, Conv3 in text prior to Figure 1. (v) The arrow connecting the five Head blocks and the subsequent parallel processes can be better.

Referring to line 68. “Additionally, many bounding boxes are created to achieve high recall.” It's a good idea to explain why high recall is more important than other performance metrics in the context of this work.

Referring to Figure 2. Explain briefly the origin of 3 by 3 kernel and the 8 by 8 matrix.

Referring to Figure 3. (i) Kindly define K, W, C, r, as it was not found anywhere in the text prior to Figure 3. Furthermore, it is a good idea to use these terms in the text when explaining the Channel Attention architecture. (ii) Kindly enlarge the arrow head. The two arrows pointing the Feature Output are barely visible.

Referring to Equation 1. It is missing one more closing bracket. Kindly check.

Referring to Figure 4. Most the feedback given for Figure 1 can be applied to Figure 4 as well. I believe the arrow for up and down sampling process are not colored.

Referring to Figure 5. Define all of the parameters involved (such as Bn, SILU, and many more) and ensure that the same parameters are used in the text for explanation and description.

Referring to Figure 6. It is preferable to arrange (a) and (b) top and bottom rather than left and right. This creates more room for the figure to be expanded horizontally.

Referring to Equation 5. Kindly define g in FocalLoss(p, g). It is a good idea to double-check that all of the equations have been properly defined.

Referring to Section 4.1 Implementation Details. (i) Kindly justify the choice of 10 epochs for 34 batches and 30 epochs for 50 batches, as these numbers are too specific. (ii) For the data augmentation technique, kindly justify the application of random crops, color variation, and rotation invariant. What are their purpose in the context of this work? It would be best if the author can show samples of the original images. Subsequently, provide justification as to why data augmentation is necessary in the first place.

Referring to line 224. “Average of average precision (AP)” can be written as “mean average precision (mAP)”, which are also defined in equation 12. Furthermore, kindly justify why mAP holds higher priority than the other performance metrics.

General feedback. Since the author has a high specification machine to conduct the experiment, it is highly recommended to at least perform cross validation on the proposed dataset and subsequently use the average and standard deviation of the mAP to represents the overall performance of the proposed model.

I wish the author all the best in their work.

Author Response

(The authors gave the same response as above.)

Round 2

Reviewer 2 Report

Dear authors, 

Thank you for all your effort. I believe my concerns has been addressed.

Many thanks.

Author Response

Dear reviewer 2

Thank you for making me reconsider the revision of the thesis.

It has helped me a lot in improving the quality of my thesis.

best regards author